# Stepwise π-extension of double [5] helicene diimides to planar nanographene diimides

Vikas Sharma[1], Jacob Isaac[2,3], Anmol Thanai[1], Kieran Richards[2,3], Daniel T. W. Toolan[4], George F. S. Whitehead [1], Emrys W. Evans [2,3] & Ashok Keerthi [1,5,6] ✉

A facile, stepwise synthetic route has been developed to access symmetric double [5]helicene diimides and their planar nanographene diimide counterparts *via* a C-shaped asymmetric [5]helicene. The synthetic strategy employs benzannulation and Scholl reaction methodologies to achieve progressive π-extension, yielding a new class of n-type rylene diimides with reversible redox characteristics. These helical and planar diimides exhibit variable crosswise π-conjugation and structural tunability, resulting in emission wavelengths that can be tailored alongside enhanced photoluminescence quantum yields—from 12% for the S-shaped diimide, to 57% for the C-shaped intermediate, and up to 63% for the fully planar nanographene diimide. Such properties make them promising candidates for quantum photonics, particularly as single-photon emitters. Photophysical properties, including time-resolved photoluminescence and transient absorption spectroscopy, reveal correlations between molecular structure, exciton dynamics, and emission behaviour. Notably, these helical rylene diimides demonstrate high photoluminescence efficiency in the solid state, reaching up to 32%, positioning them as strong contenders for next-generation optoelectronic devices.

There has been a growing fascination with helical and planar nanographenes (NGs) as optoelectronic materials in recent years[1–3]. Helical nanographenes (h-NGs) possess an inherent chirality that grants them a significant boost in cutting-edge technology applications[4,5]. These encompass a wide range of applications, such as chiral-induced spin filters, three-dimensional (3D) display, quantum communication, information storage, and encryption[6–11]. Helicene-based materials exhibit impressive absorption coefficients and significant dissymmetry factors due to their effective π-conjugation and inherent chirality. Nevertheless, the limited fluorescence quantum yield and subpar charge carrier mobility hinder its potential in advanced organic electronics[12,13]. Various methods can be used to modify the chiroptical and redox properties of basic helicenes. These include expanding the π-system along the peripheral or backbone position of the main helix core, introducing appropriate heteroatoms or functional groups in the helical core, and controlling the molecular symmetry and multiplicity of the helicene[14–22].

A wide range of synthetic strategies have been developed for the construction of polycyclic aromatic hydrocarbons (PAHs), enabling access to both helical and planar nanographenes. Helically twisted nanographenes can be generated by introducing steric hindrance or through regioselective annulation that induces axial chirality. Synthetic methods such as photocyclization, ring-closing metathesis, Diels–Alder reactions, [2 + 2 + 2] cycloisomerization, and transition metal–catalyzed cyclizations have been extensively employed to produce π-extended systems with helical topologies[4,23–25]. In contrast, planar nanographenes are often synthesized directly through oxidative cyclodehydrogenation, such as the Scholl reaction. Moreover, certain helical nanographene precursors can be transformed into planar structures under these oxidative conditions[26]. Therefore, the choice of synthetic route plays a critical role in determining the final molecular geometry and should align with the desired structural and functional properties. These advancements have expanded the possibilities for investigating chiral materials in different arrangements configurations.

[1]Department of Chemistry, School of Natural Sciences, The University of Manchester, Manchester, UK. [2]Department of Chemistry, Swansea University, Singleton Park, Swansea, UK. [3]Centre for Integrative Semiconductor Materials, Swansea University, Fabian Way, Swansea, UK. [4]Department of Materials, School of Natural Sciences, The University of Manchester, Manchester, UK. [5]Photon Science Institute, The University of Manchester, Manchester, UK. [6]National Graphene Institute, The University of Manchester, Manchester, UK. ✉e-mail: ashok.keerthi@manchester.ac.uk

Recent advancements demonstrate a strategic design approach wherein planar rylene imides scaffolds—renowned for their superior thermal stability, electron-accepting capacity, and photophysical efficacy—are structurally modified to induce helical distortion. This generates helicene-like structures that integrate the optoelectronic characteristics of rylene imides with their inherent molecular chirality[27–29]. In 2020, the Ravat group introduced the simplest helical analog of rylene imides ([n]HDI-OMe) which are classified as helicene diimides (HDIs) with different numbers of helical units (Fig. 1a)[15]. The HDIs are similar to rylene imides, but they have carbo helicene core with imide groups at both terminals of the helix and methoxy groups in the inner helix. Later, the same group reported another HDI that featured methoxy groups positioned at the outer rim of the helicene core (9,10-dimethoxy-[7] helicene diimide) which leads to distinct photophysical and electrochemical properties such as enhanced fluorescence quantum yield and lifetime (Fig. 1b)[30]. The Wurthner group recently reported two novel [n]helicenobis(naphthalimides), in which two electron-accepting naphthalimide units are connected to terminals of both the [5]helicene and [6]helicene cores (Fig. 1c)[31]. Among these two helicenes, heliceno [6]-bis(naphthalimides)

manifested impressive chiroptical characteristics and fluorescence quantum yields (up to 73%) documented for red-emitting helicenes[31]. In 2021, Hu and colleagues reported a study on double helicene diimide (DHDI), in which two imide segments were fused at the peripheral regions of the basic [4]helicene structure (Fig. 1d)[32]. Nevertheless, the existence of a helicene substructure precluded the resolution of enantiomers. Double [n]helicenes are a widely studied form of helicenes, known for their simplicity and configurational stability when the *n* value exceeds 4[4]. Nevertheless, the intriguing properties of double [5]helicene imides remain unexplored.

Our synthetic strategy for constructing rylene imide-based symmetrical double helicenes and transforming them into asymmetrical helicenes —and ultimately into planar nanographene imides—offers a powerful approach for precisely tuning optoelectronic properties. In this study, we present an efficient benzannulation-based synthetic method for the preparation of the double-[5]helicene diimide (**S-NMI**), which features two imide moieties fused to double-[5]helicene units. Furthermore, under classical Scholl reaction conditions, the double-[5]helicene diimide was selectively converted into mono-[5]helicene diimides (**C-PMI**), which subsequently underwent a facile transformation into a dibenzo-fused terrylenediimide (**DB-TDI**) (Fig. 1e). To the best of our knowledge, this represents the first successful synthetic route for generating helical rylene imides with extended π-conjugation in both the bay and K-regions[33]. Unusually, a gradual increase in fluorescence quantum yield was observed as the π-conjugation extended. The unique helical structures of double-[5] helicene diimide, *P,P*-(**S-NMI**) and *M,M*-(**S-NMI**) were confirmed through single-crystal X-ray diffraction analysis.

## Results and discussion

The synthesis of **S-NMI, C-PMI**, and planar **DB-TDI** was achieved using the synthetic approach shown in Scheme 1. Direct annulation of rylene imides is an uncommon method to accomplish "angular" fusion in the K-region. In order to proceed in this particular direction, we have devised a novel synthetic pathway and identified the suitable building block as compound **1 c**[34]. This key precursor **1c** was synthesized from naphthalene-2,6-diol in three steps (see Scheme S1): (i) regioselective bromination at the 1 and 5 positions of naphthalene-2,6-diol using N-bromosuccinimide in dry THF, yielding 81%; (ii) conversion of hydroxy groups to triflates by treatment with trifluoromethanesulfonic anhydride and pyridine in DCM, affording the corresponding bis(triflate) in 85% yield; (iii) regioselective double Sonogashira coupling of 1,5-dibromonaphthalene-2,6-diyl bis(trifluoromethanesulfonate) with trimethylsilylacetylene, yielding the key precursor **1c** in 68% yield.

Compound **1c** was subjected to a twofold Suzuki–Miyaura coupling with a naphthalene monoimide (NMI) boronic ester, using Pd(PPh₃)₄ as the catalyst and potassium phosphate as the base. This reaction afforded the bis-NMI coupled product **2** in a moderate yield of 71%. Subsequent desilylation using potassium carbonate in a THF/methanol solvent mixture at room

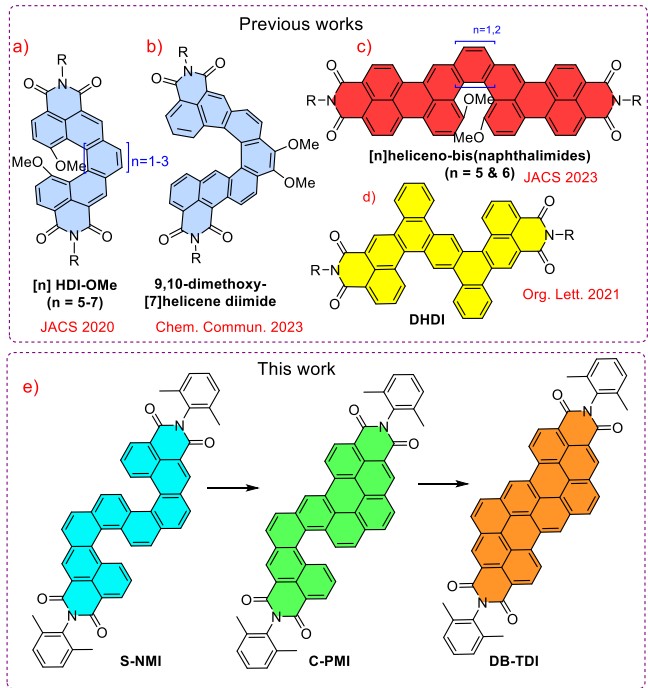

**Fig. 1 | Chemical structures of helical diimides.** Representative helical imides reported previously in the literature (**a–d**, top panel). Rylene diimides synthesised in this work (**e**, bottom panel).

**Scheme 1 | Synthesis of helical and planar rylene diimides. a** K₃PO₄, Pd(PPh₃)₄, dry toluene, ethanol, water, 100 °C, 24 h, yield 71%; **b** THF, methanol, rt, 4 h, yield 92%; **c** method 1: PtCl₂, dry toluene, 100 °C, 48 h, 52%; method 2: PtCl₂, dry toluene,

microwave reactor, 120 °C, 6 h, 70% **d**) for C-PMI: AlCl₃ (10 equivalents) chlorobenzene, 110 °C, 18 h, 55%; **e** for DB-TDI: AlCl₃ (20 equivalents) chlorobenzene, 130 °C, 48 h, 60%.

**Fig. 2 | Optoelectronic properties of rylene dimide in dilute solutions. a** Absorption spectra of **S-NMI** (cyan), **C-PMI** (green) and **DB-TDI** (orange) recorded in degassed chloroform. **b** Fluorescence spectra of **S-NMI** ($\lambda_{exc}$ = 380, cyan), **C-PMI** ($\lambda_{exc}$ = 464, green) and **DB-TDI** ($\lambda_{exc}$ = 540, orange) in degassed chloroform at room temperature.

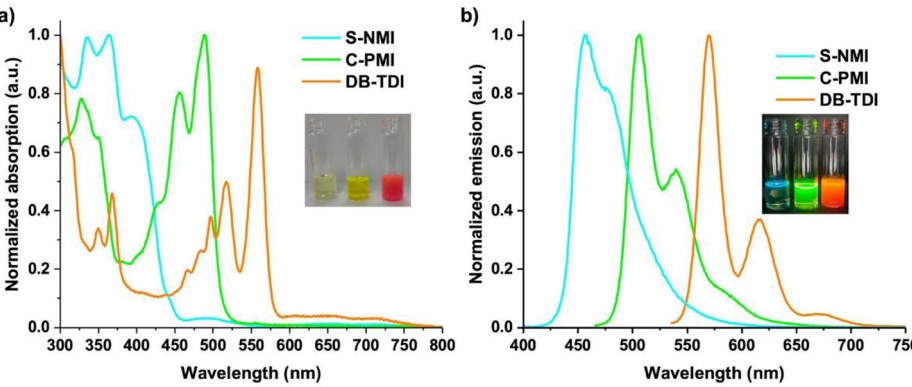

temperature for 4 h produced the bis-alkyne **3**, a key intermediate for synthesizing both helical and planar rylene diimides.

Alkyne benzannulation strategies have been widely applied in constructing non-planar and planar nanographenes, leveraging various catalysts such as transition metals, π-Lewis acids, Brønsted acids, electrophiles, and radical reagents under thermal or photochemical conditions[35]. These methods are thermodynamically favorable due to the formation of fully conjugated six-membered rings from alkyne precursors. Among these, transition metal-catalyzed benzannulations have gained significant traction, particularly due to the pioneering work of Fürstner and Storch[36–38] in developing planar and helical polycyclic aromatic hydrocarbons. Inspired by their contributions, we performed a twofold PtCl₂ catalyzed (15 mol%) alkyne benzannulation of bis-alkyne **3** in dry toluene at 80 °C for 24 h. Thin-layer chromatography analysis indicated partial conversion, with the desired double-[5]helicene diimide (**S-NMI**) formed in a modest yield of ~50%. Increasing the catalyst loading, temperature, or reaction time did not significantly improve the yield, likely due to the reduced electron density at the ipso positions of the NMI units, which hinders efficient double 6-endo-dig cyclization mediated by PtCl₂. After systematic optimization, the reaction was successfully carried out under microwave-assisted conditions using 15 mol% PtCl₂ in dry toluene at 120 °C for 6 h, affording the target double-[5]helicene diimide, **S-NMI** in a significantly improved yield of 70%.

Our objective was to sequentially and selectively form carbon–carbon bonds between naphthalene units through oxidative aryl–aryl coupling of **S-NMI**. This strategy aimed to generate π-extended **C-PMI**, an asymmetric [5]helical diimide, and **DB-TDI**, a planar nanographene diimide. However, oxidative coupling methodologies often prove ineffective for substrates with moderate electron density, necessitating the use of alternative synthetic routes. Among these, the Scholl reaction has emerged as a widely employed and powerful method for constructing both planar and helical nanographenes *via* oxidative cyclodehydrogenation[39]. Under optimized reaction conditions, **S-NMI** was heated with ten equivalents of aluminum chloride (AlCl₃) in chlorobenzene at 110 °C in a pressure tube for 18 h. This reaction resulted in the formation of an orange solid (**C-PMI**), corresponding to the selective formation of a single carbon–carbon bond between the NMI moiety and the central naphthalene core, yielding 55%. Additionally, approximately 20% of the starting materials were recovered under these conditions. Notably, increasing the amount of AlCl₃ to 20 equivalents and extending the reaction time to 48 h at 130 °C resulted in the formation of a fully conjugated planar diimide, **DB-TDI**, in 60% yield.

The UV−vis absorption and fluorescence spectroscopies have been employed to explore the photophysical characteristics of synthesized helical and planar diimides in chloroform (Fig. 2). The results obtained from experiments were further validated by theoretical calculations (see SI). The **S-NMI** exhibits three peaks with absorption maximum at 364 nm and a shoulder peak at 395 nm, which may be attributed to the electronic transition from the $S_0$ to the $S_1$ state. **C-PMI** exhibits distinct, well-defined absorption peaks in the 400–550 nm range. In contrast, weaker and less resolved peaks appear in the 300–440 nm region, likely corresponding to the

perylene and helicene subunits, respectively[31,40–42]. As the length of π-conjugation extends from **S-NMI** to **C-PMI** to **DB-TDI**, there is a significant red-shift in the absorption maxima (Fig. 2a). The absorption maxima shift progressively from 364 nm (**S-NMI**) to 490 nm (**C-PMI**) and then to 558 nm (**DB-TDI**), as shown in Fig. 2. The absorption intensity attributed to the helicene subunits decreases from **S-NMI** to **C-PMI**. This trend contrasts with previous reports on helicene–perylene bisimide hybrid molecules, where the strongest absorption peaks were attributed to the central helicene core[20,43,44]. The molar extinction coefficients for **S-NMI**, **C-PMI**, and **DB-TDI** showed a significant rise as the length of π-conjugation increased. The values obtained were 29,940 M⁻¹.cm⁻¹ (**S-NMI** at 364 nm), 63,390 M⁻¹.cm⁻¹ (**C-PMI** at 490 nm), and 68,530 M⁻¹.cm⁻¹ (**DB-TDI** at 558 nm). The optical gaps were determined from the absorption onset at higher wavelength edges are 2.8 eV for the **S-NMI**, 2.4 eV for **C-PMI**, and 2.15 eV for **DB-TDI**. This is caused by the increase in π-conjugation and increase in the energy level of the highest occupied molecular orbital (HOMO) energies as the length of the π-conjugation in the aromatic framework increases. It is expected that the lowest occupied molecular orbital (LUMO) energies are influenced by diimide moieties.

Importantly, all the helical and planar diimides show intense blue to orange fluorescence (Fig. 2b), with emission maxima that progressively redshift from 456 nm (**S-NMI**) to 505 nm (**C-PMI**), and to 569 nm (**DB-TDI**). The absolute photoluminescence quantum yields (PLQY, ϕ) of all diimides were tested in chloroform. The ϕ value obtained for **S-NMI** was 12%, and comparable with previously reported S-shaped double [4]helicene (ϕ = 13%)[32]. Interestingly, PLQY values up to 57% was achieved asymmetric [5]helicene diimide, **C-PMI** in diluted solutions of chloroform (~10⁻⁵ M). To the best of our knowledge, this is the highest reported PLQY for green light emitting [5]helical diimides. For the planar **DB-TDI**, has displayed the PLQY up to 63% in solution. Moreover, **S-NMI** maintained moderate emission in the aggregated state (3% weight in PMMA film) with a solid-state fluorescence quantum yield of 25%, which could be rationalized by the slipped and partial π − π overlaps upon aggregation between naphthalimide subunits as revealed from the single crystal analysis (Fig. 3a–c). Interestingly, **C-PMI** showed more increase in the solid-state fluorescence quantum yield up to 32% in the PMMA films. Although single crystals of C-PMI were not obtained in our experiments, the highest fluorescence quantum yield observed in thin films is likely due to a combination of two factors related to solid-state packing and interactions: (i) reduced π–π overlap between the perylene monoimide and naphthalene monoimide subunits, and (ii) π-conjugation extension in both the bay and K-regions of the perylene monoimide subunit. This result makes **C-PMI** a potential material for circularly polarized organic light-emitting diode (CP-OLED) applications. In contrast, the solid-state fluorescence quantum yield of **DB-TDI** is relatively low (up to ~9%), likely due to planar molecular structure, which facilitates strong π–π interactions between molecules.

To examine the double helical structure and intermolecular interactions in the solid state, single crystals of **S-NMI** were grown using the slow vapor diffusion of methanol into chloroform solutions. The twofold helicity

**Fig. 3 | Single-crystal X-ray diffraction and electronic circular dichroism (ECD) analysis of the enantiomers of S-NMI. a** Molecular structure of **S-NMI** determined by single-crystal X-ray diffraction; the rings along the helical core are labeled R1–R10. **b, c** Unit cell showing both enantiomers of **S-NMI**, with an inter-ring overlap distance of 3.57 Å between R1 and R1'. **d, e** ECD spectra of the **P,P**- and **M,M**-enantiomers of **S-NMI** in dichloromethane (c ≈ $10^{-5}$ M).

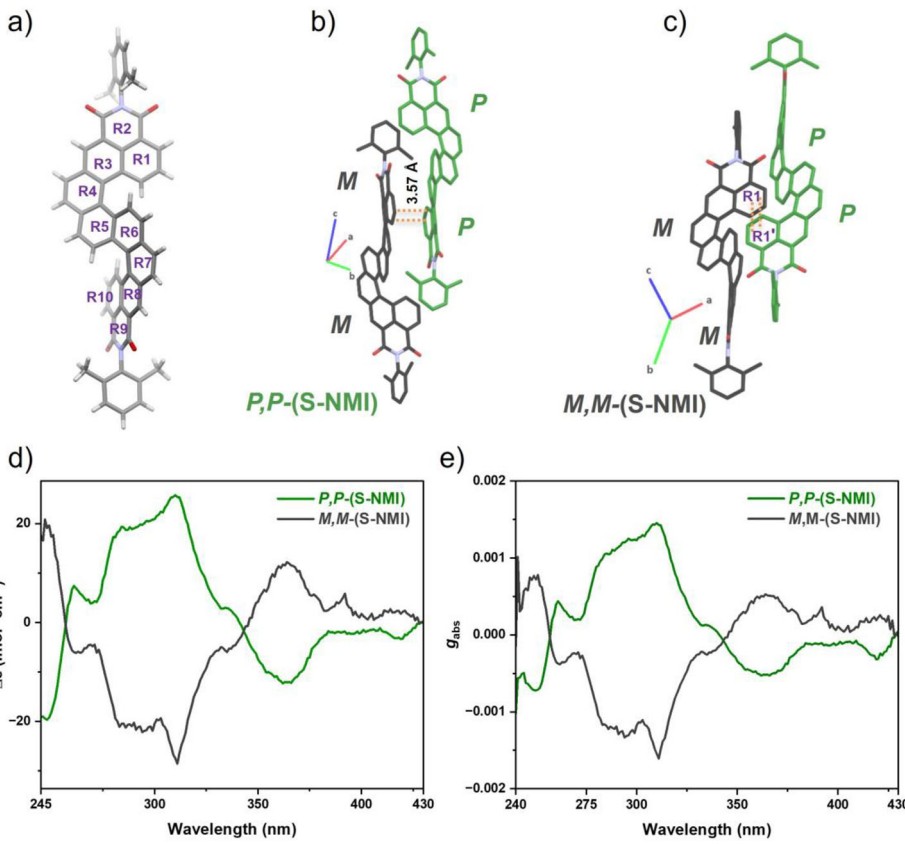

of **S-NMI** was validated using single crystal X-ray diffraction crystallography (Fig. 3a–c), which exhibits a triclinic *P-1* space group. The torsions of the two helicene subunits, characterized by the dihedral angles of the terminal rings R1/R5 and R6/R10, are 22.0° and 29.2°, respectively. Furthermore, the molecular arrangement in the crystal structure (Fig. 3b, c) exhibits a staggered π-π stacking configuration between **P,P**- and **M,M**-enantiomers of **S-NMI**. The extended crystal with an average $\pi - \pi$ stacking distance of 3.4 Å, as ascertained by the distance between the mean planes of the naphthalene monoimide subunits along the π-backbones. Prominent π–π stacking interactions are observed between the fused benzene rings R1 and R1', with a minimum centroid-to-centroid distance of 3.57 Å. However, due to the twisted molecular conformation and steric hindrance, the overall overlap between adjacent molecules is relatively limited, indicating restricted π–π interactions (see Fig. 3b).

The racemic mixtures of **S-NMI** were successfully resolved into their respective enantiomers *via* high-performance liquid chromatography (HPLC) on a chiral stationary phase LC column, Lux 5 um i-Amylose-3 (250 mm × 21.1 mm) using dichloromethane and hexane solvent mixture (90:10 ratio) as an eluent. The enantiomers of **S-NMI** were effectively isolated, and the separated enantiomers showed a Cotton effect across the 280–450 nm range, having mirror-image electronic circular dichroism (ECD) spectra in dichloromethane. The initial eluted enantiomer exhibited a positive Cotton effect at 310 nm (Δε = +26 $M^{-1}$ $cm^{-1}$), whereas the follow-up enantiomer had a negative Cotton effect of similar magnitude (Δε = –27 $M^{-1}$ $cm^{-1}$), therefore affirming their enantiomeric characters. According to our ECD spectroscopy findings and reported ECD spectra for analogous helical imides, the first eluted enantiomer has been assigned as **P,P**-(**S-NMI**), whereas the second is assigned as **M,M**-(**S-NMI**)[15,31]. Both enantiomers exhibited significant absorption dissymmetry factors ($g_{abs}$) in the UV-visible spectrum, with gabs values at 310 nm of $1.4 \times 10^{-3}$ for **P,P**-(**S-NMI**) and $1.5 \times 10^{-3}$ for **M,M**-(**S-NMI**). However, efforts to resolve other racemic mixtures for **C-PMI** were unsuccessful due to near overlapping retention times for both **M**- and **P**- enantiomers of **C-PMI** on under optimal

chiral HPLC conditions using our Lux 5 um i-Amylose-3 column with various eluent compositions. The inadequate chromatographic resolution rendering the assigning of enantiomeric ECD spectra for these samples inaccurate.

Transient photoluminescence (PL) profiles were recorded for **S-NMI**, **C-PMI** and **DB-TDI** in chloroform by time-correlated single photon counting (Fig. 4a–c). Using the equations φ = 100% × $k_r$ × τ and $k_r$ + $k_{nr}$ = 1/τ, we calculated the radiative ($k_r$) and nonradiative ($k_{nr}$) decay rate constants as shown in Table 1. Thus, the trend of PLQY can be attributed to the increase in the rate of radiative decay with more planarized structures on going from **S-NMI** ($2.1 \times 10^7\,s^{-1}$) to **C-PMI** ($2.5 \times 10^8\,s^{-1}$) and **DB-TDI** ($2.1 \times 10^8\,s^{-1}$). This is coupled with the trend of slower non-radiative decay for **DB-TDI** ($1.1 \times 10^8\,s^{-1}$) compared to **S-NMI** ($1.5 \times 10^8\,s^{-1}$) and **C-PMI** ($1.9 \times 10^8\,s^{-1}$). The fluorescence lifetimes of all three molecules are significantly shorter compared to the corresponding [n]helicenes and are more consistent with the typical values observed for rylene diimides.

In order to explore the non-radiative decay mechanism and exciton dynamics, transient absorption studies were performed for samples in chloroform (Fig. 4d–i). For **DB-TDI** (557 nm excitation), a broad photoinduced absorption (PIA) band (380–460 nm) decays with a time constant of 3.1 ± 0.07 ns at 390 nm that we attribute to the singlet exciton state ($S_1$). The assignment follows as the timescale matches the transient PL kinetics (Figs. 4a, 3.0 ns) and ground-state bleach dynamics at 370 nm (3.5 ± 0.41 ns), suggesting $S_1$ decays by emission or internal conversion to the ground state. The behavior of **DB-TDI** is similar to previously reported terrylene derivatives by us and others[45–47]. For **C-PMI** (587 nm excitation) and **S-NMI** (410 nm excitation), PIA features are found at 360–440 nm and 425–475 nm with longer lifetimes than the emission timescales: **C-PMI** (3.7 ± 0.14 ns, 375 nm) and **S-NMI** (11.6 ± 0.24 ns, 458 nm). This suggests a departure from $S_1 \rightarrow S_0 + h\nu$ only dynamics. This is further supported by longer timescale ground state bleach dynamics that do not match these kinetics: **C-PMI** (7.1 ± 0.21 ns, 465 nm) and **S-NMI** (>8 ns, 409 nm). The much longer ground state bleach feature in **S-NMI** may support unlocked

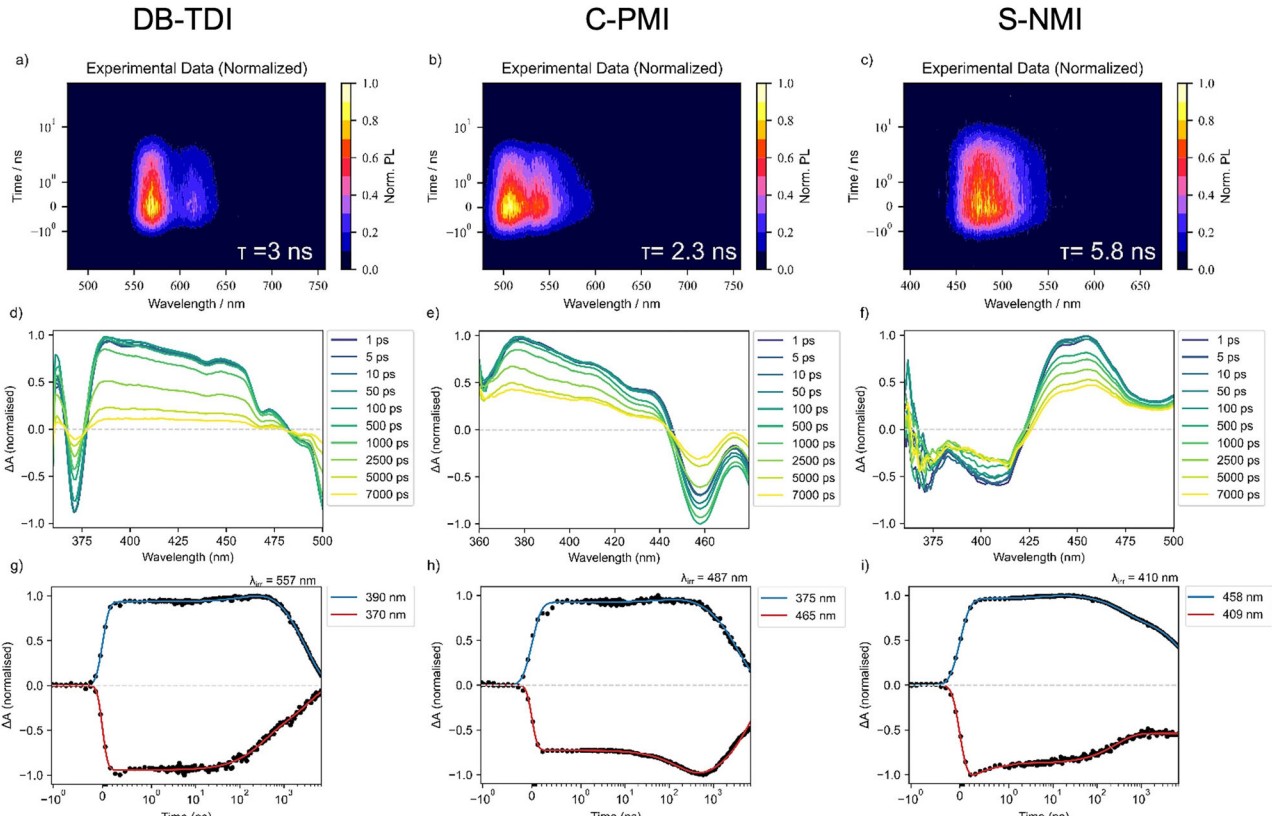

**Fig. 4 | Transient photoluminescence and transient absorption spectroscopy of helical and planar diimides.** Transient photoluminescence profiles of (**a**) **DB-TDI**; **b C-PMI**; **c S-NMI**. All samples were prepared in chloroform (0.02 mg/ml for S-NMI & C-PMI, 0.031 mg/ml for DB-TDI). Fits with convolution of instrument response function are indicated. Transient absorption spectral slices for (**d**) **DB-**

**TDI**, excitation 557 nm; **e C-PMI**, excitation 487 nm; **f S-NMI**, excitation 410 nm are shown from 1 ps to 7 ns. Corresponding kinetic profiles for photoinduced absorption (blue) and ground state bleach (red) are given for: **g DB-TDI**, 390 and 370 nm; **h C-PMI**, 375 nm and 465 nm; **i S-NMI**, 458 nm and 408 nm.

**Table 1 | Summary of experimental data from photophysical and electrochemical studies as well as DFT calculations data**

| Property measured | S-NMI | C-PMI | DB-TDI |
|---|---|---|---|
| Absorption $\lambda_{max}$ (nm) | 364 | 490 | 558 |
| Emission $\lambda_{max}$ (nm) | 456 | 505 | 569 |
| $\varepsilon$ (M$^{-1}$.cm$^{-1}$) | 29,940 | 63,390 | 68,530 |
| $\phi$ (%) | 12 | 57 | 63 |
| $\tau$ (ns) | 5.8 ($\pm$ 0.005) | 2.3 ($\pm$ 0.001) | 3.0 ($\pm$ 0.002) |
| $k_r$ (s$^{-1}$) | $2.1 \times 10^7$ | $2.5 \times 10^8$ | $2.1 \times 10^8$ |
| $k_{nr}$ (s$^{-1}$) | $1.5 \times 10^8$ | $1.9 \times 10^8$ | $1.2 \times 10^8$ |
| $E_{HOMO}$ / $E_{LUMO}$ (eV)[a] | −6.09 / −3.28 | −5.92 / −3.59 | −5.39 / −3.57 |
| $E_{HOMO}$ / $E_{LUMO}$ (eV)[b] | −6.30 / −2.75 | −6.05 / −3.17 | −5.90 / −3.44 |
| $E_g^{ec}$ (eV)[c] | 2.8 | 2.3 | 1.8 |
| $E_g^{opt}$ (eV)[d] | 2.8 | 2.4 | 2.1 |

[a]$E_{HOMO}$ = -[(Oxi$_{onset}$) + 4.8] eV; $E_{LUMO}$ = -[Red$_{onset}$) + 4.8] eV
[b]$E_{HOMO}/E_{LUMO}$ (eV) values from DFT calculations
[c]Electrochemical bang gap $E_g^{ec}$ (eV) = $E_{LUMO}$-$E_{HOMO}$
[d] $E_g^{opt}$ (eV) = 1240/$\lambda_{onset}$; where $\lambda_{onset}$ = onset of absorption in a chloroform solution)

pathways to long-lived excited states such as triplet excitons or charge transfer states that are supported by solvent polarity effects on the emission dynamics (see Figure S7-9). The results show control of not only the emission energy by modifying conjugation but also the exciton mechanism.

Rylene diimides are well-known for their redox characteristics, hence these synthesized helical and planer diimides were electrochemically examined by cyclic voltammetry (CV) in dichloromethane and chloroform,

using tetrabutylammonium hexafluorophosphate (n-Bu$_4$NPF$_6$) as a supporting electrolyte (Fig. 5 and Fig. S3). The **S-NMI** analysis revealed a reversible reduction wave at $E_{red1}$ = -1.55 V, which was followed by an additional quasi-reversible reduction wave with half-wave potentials of at $E_{red2}$ = -1.66 V, and $E_{red3}$ = -2.07 V. Interestingly, **C-PMI** showed three one-electron reduction processes with half-wave potentials at $E_{red1}$ = -1.36 V, $E_{red2}$ = -1.49 V, and $E_{red3}$ = -2.09 V. Surprisingly, the difference in reduction potentials ($\Delta E_{red}$ = $E_{red1}$ - $E_{red2}$) between the two peaks of **S-NMI** and **C-PMI** is 0.11 and 0.13 V, respectively. In contrast to rylene diimides, where the reduction potential difference ($\Delta E_{red}$) decreases with increasing $\pi$-conjugation, our system exhibits a distinctly different behavior[48]. While through-bond conjugation is the primary factor influencing rylene diimides, in our case, the close spatial proximity of the charged NMI/PMI and NMI moieties renders intramolecular through-space interactions the dominant factor[31]. The fully planar **DB-TDI** displays two quasi-reversible reduction waves at $E_{red1}$ = -1.24 V and $E_{red2}$ = -1.40 V. The complete cyclic voltammograms, including both reduction and oxidation peaks for all compounds, are provided in Figure S3.

Additionally, the HOMO and LUMO energy levels of all materials were calculated from the onset oxidation and redox potentials. The apparent increase in HOMO energy and decrease in LUMO energy during $\pi$-extension can be attributed to the enhanced electronic delocalization along the conjugated skeleton. Frontier molecular orbitals disperse over a broader region as the $\pi$-system expands (see Fig. S4). This decreases the LUMO level as a result of enhanced electron affinity and increases the HOMO level due to increased instability (more electron donation)[49]. Which reduces the electrochemical bandgap, obtained from cyclic voltammetry show good agreement with the optical band gaps derived from absorption spectra, as shown in Table 1. These patterns are clearly illustrated by the cyclic

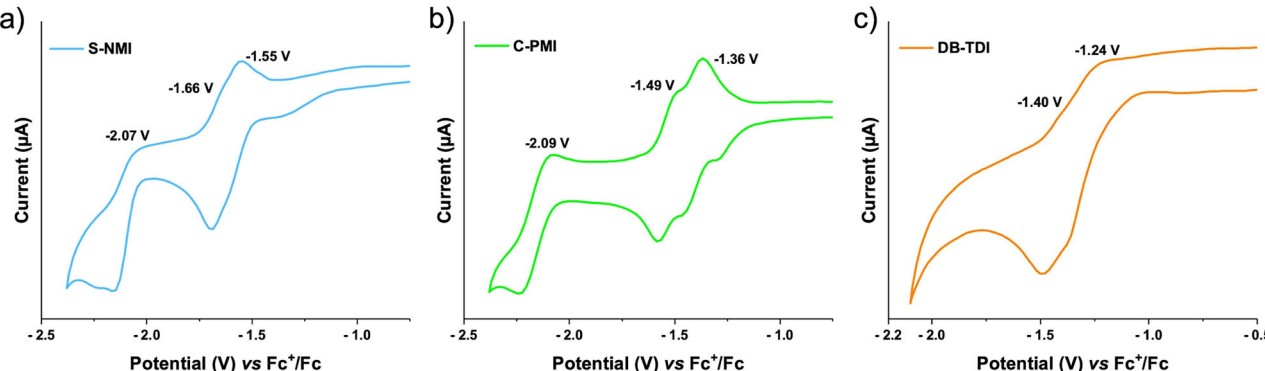

**Fig. 5 | Cyclic voltammetry studies.** Cyclic voltammograms of (**a**) **S-NMI** and **b C-PMI** were recorded in dichloromethane, and **c DB-TDI** was measured in chloroform, using 0.1 M tetrabutylammonium hexafluorophosphate as the supporting electrolyte. All measurements were performed at a scan rate of 100 mV s$^{-1}$. Potentials were calibrated against the Fc/Fc$^+$ redox couple, which served as an internal standard.

voltammetry data (Table 1), which demonstrate the direct effect of increased π-conjugation on the alignment of molecular orbitals and redox behavior. Notably, there is strong correlation between the HOMO and LUMO values estimated from density functional theory (DFT) calculations (see Fig. S4) and those determined from cyclic voltammetry measurements.

To gain further insights into the molecular packing of **S-NMI**, **C-PMI** and **DB-TDI** in the solid state, films (thin films of diimides were obtained by spin-coating 0.5 mg/mL concentration solutions in chlorobenzene) was probed *via* grazing incidence wide-angle X-ray scattering (GIWAXS) measurements (Fig. S10). Data shows that **S-NMI** and **C-PMI** films possess a high degree of molecular order as evident by the presence of Debye-Scherrer scattering rings, whilst **DB-TDI** is significantly less ordered. Scattering data for **S-NMI** and **C-PMI** exhibit larger order crystal lamellae scattering features with peaks at 0.46 & 0.55 Å$^{-1}$ (d = 13.96 & 11.43 Å), respectively. This data indicates that the additional carbon-carbon bond between NMI and the central naphthalene core from **S-NMI** > **C-PMI** results in the formation of a smaller packing motif, which is a likely consequence of increased rigidity in **C-PMI**. Furthermore, it should be noted that the film formed from **C-PMI** exhibited bright Bragg diffraction spots, indicative of presence of large, orientated crystals as opposed to the random orientation of smaller crystals as observed for **S-NMI**. The apparent decrease in molecular ordering between **C-PMI** and **DB-TDI** likely arises from the decrease in solubility and the subsequent propensity for **DB-TDI** to aggregate. Further, in-depth structural studies on of **S-NMI**, **C-PMI** and **DB-TDI** could focus on both optimising film formation conditions and solvent selection in order to maximise the molecular order of these newly synthesized materials for application in optoelectronic devices.

## Conclusions

In conclusion, we report an efficient stepwise synthetic strategy for the preparation of symmetric double-[5]helicene diimide, asymmetric [5]helicene diimide, and planar NG diimides via benzannulation and Scholl reaction approaches. The synthesized compounds, **S-NMI**, **C-PMI**, and **DB-TDI**, exhibit high fluorescence quantum yields in solution, measured at 12%, 57%, and 63%, respectively. Remarkably, both **S-NMI** and **C-PMI** retain notable solid-state fluorescence, with quantum yields of 25% and 32%, respectively addressing a longstanding limitation of planar rylenediimides. To the best of our knowledge, these represent the highest reported solid-state fluorescence quantum yields for helicene-based imides to date. Transient absorption spectroscopy was employed to probe the non-radiative decay pathways and exciton dynamics of the materials. Furthermore, single-crystal X-ray diffraction unambiguously confirmed the molecular structure of the double-[5]helicene rylenimide and enantiomers of **S-NMI** were separated on chiral HPLC. GIWAXS studies revealed that both **S-NMI** and **C-PMI** form highly ordered thin films, with small structural variations within the molecular series exerting a pronounced influence on film morphology and molecular packing. These findings underscore the significant potential of the newly synthesized materials for application in advanced optoelectronic devices, particularly circularly polarized organic light-emitting diodes.

## Materials and methods
### Materials
All reagents were purchased from commercial sources and used without further purification unless otherwise stated. Thin layer chromatography was carried out using pre-coated aluminum sheets with silica gel 60 F254 (Merck). Column chromatography was performed using Merck silica gel (60 Å, 230–400 mesh).

### General methods
$^1$H-NMR and $^{13}$C-NMR spectra were recorded in the listed deuterated solvents (CDCl$_3$, 1,1,2,2-Tetrachloroethane-d$_2$) on a Bruker 400, and 500 MHz spectrometers. Atmospheric Pressure Chemical Ionization (APCI) was performed using a Thermo Orbitrap Exactive Plus Extended Mass Range mass spectrometer.

### Crystallography
X-ray diffraction data for compound S-NMI were collected using a dual wavelength Rigaku FR X rotating anode diffractometer using CuKα (λ = 1.54146 Å) radiation, equipped with an AFC 11 4 circle goniometer, VariMAXTM microfocus optics, a Hypix-6000HE detector and an Oxford Cryosystems 800 plus nitrogen flow gas system, at a temperature of 100 K. Data were collected and reduced using CrysAlisPro v43[50]. Absorption correction was performed using empirical methods (SCALE3 ABSPACK) based upon symmetry-equivalent reflections combined with measurements at different azimuthal angles.

Crystal structure determination and refinements: The crystal structure was solved and refined against all F2 values using the SHELX and Olex2 suite of programs[51,52]. Coordinates and anisotropic displacement parameters for all non-hydrogen atoms were freely refined. Hydrogen atoms were constrained to idealised positions with the coordinates refined to ride with the parent atom. Hydrogen isotropic atomic displacement parameters were constrained to ride with the parent atom with an appropriate multiplier for the hybridisation

### Grazing incidence wide-angle X-ray scattering (GIWAXS) measurements
GIWAXS measurements were performed using a Xeuss 3.0 laboratory beamling (Xenocs) equipped with a liquid gallium MetalJet source (Excillum), producing X-rays with an energy of 9.243 keV (λ = 1.34 Å). A collimated X-ray beam was directed at sample surfaces inclined at a grazing angle of 0.15° and scattered X-rays were detected by a Pilatus3R 1 M 2D

X-ray detector (Dectris) positioned ~300 mm from the sample center. The sample to detector distance was calibrated using a silver behenate standard in transmission geometry. GIWAXS data were corrected, reshaped and reduced using code based on pyfai and pygix python libraries. 1D intensity profiles were generated by azimuthally integrating the 2D patterns as a function of q = $4\pi\sin\theta/\lambda$ where $2\theta$ is the angle between the incident and scattered X-ray of wavelength $\lambda$. Integrations were performed across the full q range.

## Preparative HPLC

Chiral resolution of S-NMI conducted on a preparative HPLC system, Agilent Technologies 1260 Infinity instrument with a Lux 5 μm i-Amylose-3 LC column (250 mm × 21.1 mm). Eluent for S-NMI: $CH_2Cl_2$/n-hexane = 90/10 (v/v), C-PMI: $CH_2Cl_2$/n-hexane = 90/20 (v/v),15 mL/min. The detection wavelength was 254 nm.

## Spectroscopy measurements

UV-vis absorption spectra and optical densities were measured using a Varian Cary 5000 UV-Vis-NIR spectrophotometer, while fluorescence emission spectra were gathered using a Varian Cary Eclipse fluorescence spectrophotometer at 298 K. Time-resolved photoluminescence profiles were recorded using a Hamamatsu C10910 Streak Camera with laser excitation provided by a Light Conversion PHAROS pumped ORPHEUS OPA at 10 kHz. The photoluminescence quantum yield (PLQE) of solutions were measured using a Quantaurus-QY® Plus UV-NIR absolute PL quantum yield spectrometer (C13534 series) and a Xe Lamp.

Transient Absorption (TA) Measurements were carried out using a commercial fsTA setup (HARPIA-TA, Light Conversion). utilising a 1030 nm Yb:KGW 25 kHz laser source (Pharos, Light Conversion). Pump excitation was generated using a non-linear optical parametric amplifier (OPA) capable of generating femtosecond laser pulses between 300–2700 nm (Orpheus + Lyra, Light Conversion). UV (360–500 nm) probe pulses were generated through irradiation of a sapphire crystal with 515 nm radiation. A 50 Hz chopper was used and the temporal resolution is achieved using a delay stage with a maximum delay of 8 ns.

## Data availability

CCDC 2451837 contain the supplementary crystallographic data for this paper. These data can be obtained free of charge via https://www.ccdc.cam.ac.uk/structures/, or by emailing data_request@ccdc.cam.ac.uk, or by contacting The Cambridge Crystallographic Data Center, 12 Union Road, Cambridge, CB2 1EZ, UK; fax: +44 1223 336033. Supporting Information contains experimental details, NMR spectroscopic analysis, HR-MS data, single crystal X-ray structures, CV spectra, DFT calculations and GIWAXS data. All the other data supporting the findings of this study are stored on our institutional server and are available from the corresponding author upon reasonable request.

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

## Acknowledgements
A.K. acknowledge EPSRC new horizon award EP/V048112/1 and EPSRC strategic equipment grant EP/W006502/1. This work is partially supported by Royal Society grant ICA\R1\231014. A.K. acknowledges Presidential Fellowship from the University of Manchester. We are thankful to the support from NMR team and the Mass Spectrometry & Separations Facility team as well as helpful discussions with Dr Avantika Hasija at XRD facility in the Department of Chemistry at the University of Manchester. We acknowledge Diamond Light Source for access to the DL-SAXS equipment (experiment number SM40538-1) supported by an EPSRC grant (EP/R042683/1), and instrument scientist Dr Paul Wady for their help and support during beam time.

## Author contributions
A.K. and V.S. designed the project. V.S. synthesized the compounds, and carried out their characterization using NMR spectroscopy, HRMS, and optoelectronic studies (UV–vis spectroscopy, fluorescence spectroscopy, and cyclic voltammetry). J.I. and K.R. performed transient photoluminescence and transient absorption spectroscopy. A.T. separated the S-NMI isomers on chiral preparative HPLC and conducted CD spectroscopy. D.T.W.T. performed GIWAXS measurements. G.F.S.W. carried out crystallographic analysis. E.W.E. supervised J.I and K.R. on the transient spectroscopy measurements and data analysis. A.K. performed DFT calculations and supervised the project. All authors contributed to the preparation of the manuscript.

## Competing interests
The authors declare no competing interests
