## [Transparent Peer Review file · Communications Chemistry]

Stepwise π -Extension of double [5]Helicene Diimides to Planar Nanographene Diimides

Corresponding Author: Dr Ashok Keerthi

Version 0:

Reviewer comments:

Reviewer #1

(Remarks to the Author)
See attached.

Reviewer #2

(Remarks to the Author)

In this work, the authors report an efficient, stepwise synthetic strategy to access a series of n type rylene diimides: a symmetric double [5]helicene diimide (S NMI), an asymmetric [5]helicene diimide (C PMI), and a fully planar nanographene diimide (DB TDI). These compounds are characterized by X ray crystallography, UV–Vis absorption and time resolved photoluminescence spectroscopy, transient absorption spectroscopy, cyclic voltammetry, and GIWAXS thin film measurements. The authors demonstrate a progressive red shift of both absorption and emission maxima, accompanied by a remarkable increase in solution photoluminescence quantum yield. These results position the new materials as promising candidates for quantum photonics and circularly polarized OLED applications. Overall, the quality of this work is very high and it is suitable for publication; however, the following comments should be addressed:

1. Given the emphasis on helicity, circular dichroism (CD) spectra for S NMI and C PMI are currently missing. The authors should include CD measurements (and dissymmetry factors) to support their claims regarding helical chirality.
2. Transient absorption reveals long lived bleach signals for S NMI and C PMI, which may indicate triplet or charge transfer states. Additional experiments—such as nanosecond transient absorption or quenching studies—could help clarify these excited state pathways.
3. In line 154 (caption of Figure 2), correct “grean” to “green.”
4. The observation that π extension raises the HOMO energy while lowering the LUMO energy (Table 1) needs further discussion. The authors should comment on how increased conjugation influences orbital energies and electrochemical behavior.

Version 1:

Reviewer comments:

Reviewer #1

(Remarks to the Author)

In their rebutal letter, the authors adressed all questions regarding content and format. Figure 1 was revised and a more consistent terminology for the description of helical and planar nanographenes is now available. With the changes made the article can be published.

Reviewer #2

(Remarks to the Author)

In the revised manuscript, the authors have effectively incorporated all comments and addressed all concerns. I have no further remarks, and the manuscript can be accepted in its current form.

We sincerely thank both reviewers for their time and effort in reviewing our manuscript and for their valuable suggestions to improve it. We are pleased to hear that both reviewers appreciated our work and recognised its relevance to the field. We have carefully considered all the comments and have significantly revised the manuscript accordingly (revised manuscript with yellow highlights). Below, we provide a point-by-point response to the reviewers' comments.

Reviewers 1

Manuscript Title: Stepwise π -Extension of double [5]Helicene Diimides to Planar Nanographene Diimides

Authors: Vikas Sharma et al.

Recommendation: Accept

1. Summary

The manuscript reports the synthetic route to double [5]helicene diimides in their core structure. The authors employ a clever strategy which involves cross-coupling of a functionalized naphthalene imide to a naphthalene core, followed by a platinum chloride-induced angular fusion in the K-region to generate the target motif. To the best of my knowledge only one synthetic methodology describes the realization of the pristine carbon-based double [5]helicene structure (Bulletin of the Chemical Society of Japan, Volume 91, Issue 7, July 2018, Pages 1069–1074, <https://doi.org/10.1246/bcsj.20180081>) but is relying on an iridium catalyzed annulation reaction.

Another claim of the authors is, that they could overcome limitations of classical oxidative cyclodehydrogenation (typically using FeCl_3) by employing AlCl_3 in chlorobenzene at elevated temperatures, achieving full cyclization and planarization of both helical centers to yield the fully planarized structure DBTDI.

Additionally, the authors report high photoluminescence quantum yields (PLQY), with values up to 57% in solution and up to 32% in the solid state— which are claimed to be the highest reported for green light-emitting helical diamides.

2. Scientific Validity

The synthetic methodology is well-designed and clearly described. Structural assignments are convincingly supported by single-crystal X-ray diffraction. Photophysical properties are thoroughly characterized and discussed in the context of existing literature. The claims are well-supported by the data presented.

3. Significance and Novelty

The work is novel and represents an advance in the synthesis of helical and planar nanographene diimides. While the main building blocks and synthetic steps have been previously described, it has not been explored in this context. The angular fusion strategy and the use of AlCl_3 for full planarization are particularly noteworthy. The reported PLQY values are impressive and address a long-standing limitation in the field according to the authors.

4. Clarity and Presentation

The manuscript is generally well-written and clearly structured. However, the first figure in the introduction could be better aligned with the molecules discussed in the text to improve clarity. Additionally, the emphasis in the introduction appears to be more on planarization, whereas the true novelty lies in the generation of the helical structures. A slight rebalancing of this focus would enhance the narrative.

5. Specific Comments

Consider revising the first figure and its placement in the introduction to better reflect the molecular structures discussed. A clearer emphasis on the novelty of the helical structure formation (rather than just the planarization) would strengthen the manuscript's impact.

We sincerely thank the reviewer for these thoughtful, constructive and encouraging feedback, as well as for recognizing the novelty and significance of our work.

In response to the suggestion regarding Figure 1 and its alignment with the text, we have revised the figure to more clearly represent the helicene imide structures discussed in the manuscript. Each molecule in the figure has now been appropriately labelled and described to facilitate better alignment with the corresponding discussion in the text.

Furthermore, we have substantially revised the introduction to more clearly highlight the novelty of our approach—specifically, the generation of double-[5]helicene diimide architectures. We have expanded the discussion on commonly employed synthetic strategies for helicene-based polycyclic aromatic hydrocarbons and emphasized how our design overcomes these limitations through a unique angular fusion strategy. This reorganization helps to better frame the significance of the helical structures in the context of our findings, while maintaining appropriate attention to the subsequent planarization. To further enrich the context and reinforce the impact of our research, we have also added additional references to key literature on helicene PAHs.

These changes are reflected in the revised Introduction section, and the updated Figure 1 is provided below for clarity.

Figure 1. Chemical structures of helical imides reported in the literature (top panel) and synthesised rylene diimides in this work.

Minor suggestions:

- 47: “Various synthetic strategies have been developed *to create h-NGs and planar structures*, such as photocyclizations, Scholl reactions, and cyclization’s catalysed by transition metals.”

I would advise to rewrite this sentence for clarity. I understand that there are synthetic strategies for the synthesis of polycyclic aromatic hydrocarbons in more general terms; or nano graphenes, independent if there are planar or helical. If the authors would like to refer to transformation of helical nanographenes into planar nanographenes by these synthetic strategies, it should be emphasized.

We sincerely thank the reviewer for their constructive suggestion. In response, we have revised the text to clearly distinguish between the synthetic strategies employed for the construction of helical versus planar nanographenes. The revised paragraph now outlines these methodologies separately, with an emphasis on the transformation from helical to planar nanographenes where applicable. This distinction better reflects the structural diversity and synthetic intent within the field.

The revised paragraph is given below:

helicene.¹⁴⁻²² A wide range of synthetic strategies have been developed for the construction of polycyclic aromatic hydrocarbons (PAHs), enabling access to both helical and planar nanographenes. Helically twisted nanographenes can be generated by introducing steric hindrance or through regioselective annulation that induces axial chirality. Synthetic methods such as photocyclization, ring-closing metathesis, Diels–Alder reactions, [2 + 2 + 2] cycloisomerization, and transition metal–catalyzed cyclizations have been extensively employed to produce π -extended systems with helical topologies.^{4,23-25} In contrast, planar nanographenes are often synthesized directly through oxidative cyclodehydrogenation, such as the Scholl reaction. Moreover, certain helical nanographene precursors can be transformed into planar structures under these oxidative conditions.²⁶ Therefore, the choice of synthetic route plays a critical role in determining the final molecular geometry and should align with the desired structural and functional properties. These advancements have expanded the possibilities for investigating chiral materials in different arrangements configurations.

- 52: “In recent years, there have been several reports of incorporating six-membered dicarboximide groups on periphery a helicene core, leveraging inspiration from the outstanding optoelectronic properties linked to rylene imides.”

My suggestion is to revise this sentence to highlight the design approach using rylene imides as starting points and converting them into helical structures as it is outlined in the sentences afterwards.

We thank the reviewer for the insightful suggestion. In response, we have revised the sentence to more clearly reflect the design strategy, which begins with planar rylene imide scaffolds and introduces helical distortion to yield helicene-like structures. The revised text now better highlights the conceptual development from rylene cores to helically contorted architectures.

Revised sentence:

Recent advancements demonstrate a strategic design approach wherein planar rylene imides scaffolds—renowned for their superior thermal stability, electron-accepting capacity, and photophysical efficacy—are structurally modified to induce helical distortion. This generates helicene-like structures that integrate the optoelectronic characteristics of rylene imides with their inherent molecular chirality.²⁷⁻²⁹ In 2020, the Ravat group introduced the simplest helical analogue of rylene imides ([*n*]HDI-OMe) which are classified as helicene diimides (HDIs) with different numbers of helical units (Figure 1a).¹⁵ The HDIs are similar to rylene imides, but they have carbo helicene core with imide groups at both terminals of the helix and methoxy groups in the inner helix. Later, the same group reported another HDI that featured methoxy groups positioned at the outer rim of the helicene core (9,10-dimethoxy-[7]helicene diimide) which leads to distinct photophysical and electrochemical properties such as enhanced fluorescence quantum yield and lifetime (Figure 1b).³⁰ The Wurthner group recently reported two novel [*n*]heliceno-bis(naphthalimides), in which two electron-accepting naphthalimide units are connected to terminals of both the [5]helicene and [6]helicene cores (Figure 1c).³¹ Among these two helicenes, heliceno [6]-bis(naphthalimides) manifested impressive chiroptical characteristics and fluorescence quantum yields (up to 73%) documented for red-emitting helicenes.³¹ In 2021, Hu and colleagues reported a study on double helicene diimide (DHDI), in which two imide segments were fused at the peripheral regions of the basic [4]helicene structure (Figure 1d).³² Nevertheless, the existence of a helicene substructure precluded the resolution of enantiomers. Double [*n*]helicenes are a widely studied form of helicenes, known for their simplicity and configurational stability when the *n* value exceeds 4.⁴ Nevertheless, the intriguing properties of double [5]helicene imides remain unexplored.

Ashok Keerthi (ashok.keerthi@r

“52-71 and Figure 1”

Please number the structures in Figure 1 and refer to them in the text, for clarity and convenience of the reader.

Thanks to reviewer for this and above suggestion. To enhance clarity and improve the readability of the manuscript, we have now numbered all structures in Figure 1 and refer to them consistently throughout the text. This adjustment allows for more precise cross-referencing and facilitates easier navigation for the reader.

- 106 “dry toluene, *uW*, 120 °C, 6h, 70%“

The abbreviation *uW* is not entirely clear to me from that context. It may refer to microwave.

To ensure clarity and ambiguity, we have replaced the abbreviation “uW” with the full term “microwave reactor” throughout the manuscript and SI to explicitly indicate the use of microwave irradiation.

6. Recommendation

Accept

The manuscript presents a novel and well-executed study that will be of interest to researchers in organic electronics, dye chemistry, and polycyclic aromatic hydrocarbons. The claims are well-supported, and the work meets the standards of Communications Chemistry.

We thank the reviewer for their appreciation of our and detailed report.

Reviewers 2

In this work, the authors report an efficient, stepwise synthetic strategy to access a series of n type rylene diimides: a symmetric double [5]helicene diimide (S NMI), an asymmetric [5]helicene diimide (C PMI), and a fully planar nanographene diimide (DB TDI). These compounds are characterized by X ray crystallography, UV–Vis absorption and time resolved photoluminescence spectroscopy, transient absorption spectroscopy, cyclic voltammetry, and GIWAXS thin film measurements. The authors demonstrate a progressive red shift of both absorption and emission maxima, accompanied by a remarkable increase in solution photoluminescence quantum yield. These results position the new materials as promising candidates for quantum photonics and circularly polarized OLED applications. Overall, the quality of this work is very high, and it is suitable for publication; however, the following comments should be addressed:

1. Given the emphasis on helicity, circular dichroism (CD) spectra for S NMI and C PMI are currently missing. The authors should include CD measurements (and dissymmetry factors) to support their claims regarding helical chirality.

We sincerely thank the reviewer for this insightful comment. We agree that circular dichroism (CD) measurements are crucial to support our claims regarding the helical chirality of S-NMI and C-PMI. In response, we have included CD spectra and dissymmetry factor (g_{abs}) values for the enantiomers of S-NMI, which were successfully resolved *via* preparative chiral HPLC using Phenomenex Lux 5 μm i-Amylose-3 LC column (250 mm \times 21.1 mm). These results confirm the presence of enantiomeric helical structures and are now included in the revised manuscript as Figure 2c-d. Additionally, the chiral HPLC chromatograms of both compounds have been added to the Supporting Information (Figure S7).

However, despite extensive optimization using our chiral separations column, Phenomenex Lux 5 μm i-Amylose-3 LC column (250 mm \times 21.1 mm), the enantiomeric resolution of C-PMI was unsuccessful due to overlapping retention times, preventing reliable CD spectral assignment. We have revised the manuscript accordingly to reflect these additions and limitations.

“To examine the double helical structure and intermolecular interactions in the solid state, single crystals of S-NMI were grown using the slow vapour diffusion of methanol into chloroform solutions. The two-fold helicity of S-NMI was validated using single crystal X-ray diffraction crystallography

(Figure 3a-c), which exhibits a triclinic *P-1* space group. The torsions of the two helicene subunits, characterized by the dihedral angles of the terminal rings R1/R5 and R6/R10, are 22.0° and 29.2°, respectively. Furthermore, the molecular arrangement in the crystal structure (Figure 3b and 3c) exhibits a staggered π - π stacking configuration between *P,P*- and *M,M*-enantiomers of *S-NMI*. The extended crystal with an average π - π stacking distance of 3.4 Å, as ascertained by the distance between the mean planes of the naphthalene monoimide subunits along the π -backbones. Prominent π - π stacking interactions are observed between the fused benzene rings R1 and R1', with a minimum centroid-to-centroid distance of 3.57 Å. However, due to the twisted molecular conformation and steric hindrance, the overall overlap between adjacent molecules is relatively limited, indicating restricted π - π interactions (see Figure 3b).

The racemic mixtures of *S-NMI* were successfully resolved into their respective enantiomers *via* high-performance liquid chromatography (HPLC) on a chiral stationary phase LC column, Lux 5 μ m i-Amylose-3 (250 mm \times 21.1 mm) using dichloromethane and hexane solvent mixture (90:10 ratio) as an eluent. The enantiomers of *S-NMI* were effectively isolated, and the separated enantiomers showed a Cotton effect across the 280-450 nm range, having mirror-image electronic circular dichroism (ECD) spectra in dichloromethane. The initial eluted enantiomer exhibited a positive Cotton effect at 310 nm ($\Delta\epsilon = +26 \text{ M}^{-1} \text{ cm}^{-1}$), whereas the follow-up enantiomer had a negative Cotton effect of similar magnitude ($\Delta\epsilon = -27 \text{ M}^{-1} \text{ cm}^{-1}$), therefore affirming their enantiomeric characters. According to our ECD spectroscopy findings and reported ECD spectra for analogous helical imides, the first eluted enantiomer has been assigned as *P,P*-(*S-NMI*), whereas the second is assigned as *M,M*-(*S-NMI*).^{15,31} Both enantiomers exhibited significant absorption dissymmetry factors (g_{abs}) in the UV-visible spectrum, with g_{abs} values at 310 nm of 1.4×10^{-3} for *P,P*-(*S-NMI*) and 1.5×10^{-3} for *M,M*-(*S-NMI*). However, efforts to resolve other racemic mixtures for *C-PMI* were unsuccessful due to near overlapping retention times for both *M*- and *P*- enantiomers of *C-PMI* on under optimal chiral HPLC conditions using our Lux 5 μ m i-Amylose-3 column with various eluent compositions. The inadequate chromatographic resolution rendering the assigning of enantiomeric ECD spectra for these samples inaccurate.”

Figure S11. Chromatograms of a) C-PMI (racemates); b) S-NMI (racemates); c) *P,P*-(S-NMI) and d) *M,M*-(S-NMI). Chiral separations were conducted using Lux 5 μ m i-Amylose-3 LC column (250 mm \times 21.1 mm). Eluent for S-NMI: CH₂Cl₂/n-hexane = 90/10 (v/v), C-PMI: CH₂Cl₂/n-hexane = 90/20 (v/v), with 15 mL/min flowrate.

Figure 3. Single-crystal X-ray diffraction and electronic circular dichroism (ECD) analysis of the enantiomers of S-NMI. (a) Molecular structure of S-NMI determined by single-crystal X-ray diffraction; the rings along the helical core are labeled R1–R10. (b, c) Unit cell showing both enantiomers of S-NMI, with an inter-ring overlap distance of 3.57 Å between R1 and R1'. (d, e) ECD spectra of the *P,P*- and *M,M*-enantiomers of S-NMI in dichloromethane ($c \approx 10^{-5}$ M).

2. Transient absorption reveals long lived bleach signals for S-NMI and C-PMI, which may indicate triplet or charge transfer states. Additional experiments—such as nanosecond transient absorption or quenching studies—could help clarify these excited state pathways.

We thank the reviewer for the comment and agree it desirable to give further insights into the nature of the exciton mechanism for the series. The reviewer suggests nanosecond transient absorption experiment and quenching experiment (with molecular oxygen, ³O₂) that is sensitive to triplets. We note that triplet charge transfer (³CT) states (formed by spin mixing with singlet ¹CT formed after charge transfer from S₁) and triplet local exciton states (formed by intersystem crossing from S₁) to both show reduced lifetime of the excited state in the presence of ³O₂ that may not distinguished.

We performed further time-resolved PL experiments for the series in toluene as less polar solvent compared to experiments in more polar chloroform solvent as presented. The experiments were

performed with a streak camera that enables the spectral profile to be followed over nanoseconds for **DB-TDI**, **C-PMI**, and **S-NMI**. In **DB-TDI**, we observe no change in lifetime (3 ns). With increasing solvent polarity, for **C-PMI** there is a minor reduction in lifetime (2.6 to 2.3 ns), which is most prominent in **S-NMI** (7.2 to 5.8 ns). We believe these results point to intersystem crossing between local excited S_1 and T_1 states in fully fused and rigid for **DB-TDI**, due to functional states that are not energetically affected by changing solvent polarity. In more twisted structures, a reduced lifetime occurs and may be attributed to increasing decay pathway *via* CT state that is stabilized relative to S_1 in more polar solvent and competes with S_1 emission and intersystem crossing. The absence of substantial red-shifting of the emission in these molecules also support that the functional emitting state has local excited S_1 character.

From these further experiments, we have revised the manuscript as follows.

In the manuscript: Figure 4 was updated to include streak camera data.

Figure 4. Transient photoluminescence and transient absorption spectroscopy of helical and planar diimides. Transient photoluminescence profiles of (a) **DB-TDI**; (b) **C-PMI**; (c) **S-NMI**. All samples were prepared in chloroform (0.02 mg/ml for S-NMI & C-PMI, 0.031 mg/ml for DB-TDI). Fits with convolution of instrument response function are indicated. Transient absorption spectral slices for (d) **DB-TDI**, excitation 557 nm; (e) **C-PMI**, excitation 487 nm; (f) **S-NMI**, excitation 410 nm are shown from 1 ps to 7 ns. Corresponding kinetic profiles for photoinduced absorption (blue) and ground state bleach (red) are given for: (g) **DB-TDI**, 390 and 370 nm; (h) **C-PMI**, 375 nm and 465 nm; (i) **S-NMI**, 458 nm and 408 nm.

Transient photoluminescence (PL) profiles were recorded for **S-NMI**, **C-PMI** and **DB-TDI** in chloroform by time-correlated single photon counting (Figure 4a-c). Using the equations $\phi = 100\% \times k_r \times \tau$ and $k_r + k_{nr} = 1/\tau$, we calculated the radiative (k_r) and nonradiative (k_{nr}) decay rate constants as shown in Table 1. Thus, the trend of PLQY can be attributed to the increase in the rate of radiative decay with more planarized structures on going from **S-NMI** ($2.1 \times 10^7 \text{ s}^{-1}$) to **C-PMI** ($2.5 \times 10^8 \text{ s}^{-1}$) and **DB-TDI** ($2.1 \times 10^8 \text{ s}^{-1}$). This is coupled with the trend of slower non-radiative decay for **DB-TDI** ($1.1 \times 10^8 \text{ s}^{-1}$) compared to **S-NMI** ($1.5 \times 10^8 \text{ s}^{-1}$) and **C-PMI** ($1.9 \times 10^8 \text{ s}^{-1}$). The fluorescence lifetimes of all three molecules are significantly shorter compared to the corresponding [n]helicenes and are more consistent with the typical values observed for rylene diimides.

In the SI:

7. Time-resolved photoluminescence data

Figure S7. Global analysis of time-resolved photoluminescence data in both toluene and chloroform for **DB-TDI**. **a, d)** Experimental 2D time-resolved emission plots. **b, e)** Corresponding fitted data obtained from global analysis. **c, f)** Decay-associated spectra (DAS) extracted from the fits, showing distinct spectral components and their associated lifetimes (τ).

Figure S8. Global analysis of time-resolved photoluminescence data in both toluene and chloroform for **C-PMI**. **a, d)** Experimental 2D time-resolved emission plots. **b, e)** Corresponding fitted data obtained from global analysis. **c, f)** Decay-associated spectra (DAS) extracted from the fits, showing distinct spectral components and their associated lifetimes (τ).

S-NMI in Chloroform

Figure S9. Global analysis of time-resolved photoluminescence data in both toluene and chloroform for **S-NMI**. **a, d)** Experimental 2D time-resolved emission plots. **b, e)** Corresponding fitted data obtained from global analysis. **c, f)** Decay-associated spectra (DAS) extracted from the fits, showing distinct spectral components and their associated lifetimes (τ).

0.24 ns, 458 nm). This suggests a departure from $S_1 \rightarrow S_0 + h\nu$ only dynamics. This is further supported by longer timescale ground state bleach dynamics that do not match these kinetics: **C-PMI** (7.1 ± 0.21 ns, 465 nm) and **S-NMI** (> 8 ns, 409 nm). The much longer ground state bleach feature in **S-NMI** may support unlocked pathways to long-lived excited states such as triplet excitons or charge transfer states that are supported by solvent polarity effects on the emission dynamics (see Figure S7-9). The results show control of not only the emission energy by modifying conjugation but also the exciton mechanism.

3. In line 154 (caption of Figure 2), correct “grean” to “green.”

We are grateful to the reviewer for pointing the typo. Figure 2 caption is corrected.

4. The observation that π extension raises the HOMO energy while lowering the LUMO energy (Table 1) needs further discussion. The authors should comment on how increased conjugation influences orbital energies and electrochemical behavior.

We thank the reviewer for this important comment. In response, we have expanded the discussion in the revised manuscript to better contextualize the impact of π -extension on the frontier molecular orbital energies and electrochemical properties. As shown in Table 1, π -extension leads to a notable elevation of the HOMO energy and a concurrent lowering of the LUMO energy. This trend can be rationalized by considering that increased conjugation enhances electron delocalization across the molecular backbone, which in turn stabilizes the LUMO through extended π^* -orbital overlap while destabilizing the HOMO due to greater electron density in bonding orbitals.

From an electrochemical perspective, this narrowing of the HOMO–LUMO gap is reflected in a decreased electrochemical bandgap and a shift of oxidation and reduction potentials toward more accessible values. These effects underscore the role of π -extension not only in modulating optical transitions but also in fine-tuning redox behavior—features particularly desirable for applications in organic electronics and energy conversion. The relevant discussion has been incorporated into the revised manuscript (page 13).

Additionally, the HOMO and LUMO energy levels of all materials were calculated from the onset oxidation and redox potentials. The apparent increase in HOMO energy and decrease in LUMO energy during π -extension can be attributed to the enhanced electronic delocalization along the conjugated skeleton. Frontier molecular orbitals disperse over a broader region as the π -system expands (see Figure S4). This decreases the LUMO level as a result of enhanced electron affinity and increases the HOMO level due to increased instability (more electron donation).⁴⁹ Which reduces the electrochemical bandgap, obtained from cyclic voltammetry show good agreement with the optical band gaps derived from absorption spectra, as shown in Table 1. These patterns are clearly illustrated by the cyclic voltammetry data (Table 1), which demonstrate the direct effect of increased π -conjugation on the alignment of molecular orbitals and redox behavior. Notably, there is strong correlation between the HOMO and LUMO values estimated from density functional theory (DFT) calculations (see Figure S4) and those determined from cyclic voltammetry measurements.

Manuscript Title: Stepwise π -Extension of double [5]Helicene Diimides to Planar Nanographene Diimides

Authors: Vikas Sharma et al.

Recommendation: Accept

1. Summary

The manuscript reports the synthetic route to double [5]helicene diimides in their core structure. The authors employ a clever strategy which involves cross-coupling of a functionalized naphthalene imide to a naphthalene core, followed by a platinum chloride-induced angular fusion in the K-region to generate the target motif. To the best of my knowledge only one synthetic methodology describes the realization of the pristine carbon-based double [5]helicene structure (Bulletin of the Chemical Society of Japan, Volume 91, Issue 7, July 2018, Pages 1069–1074, <https://doi.org/10.1246/bcsj.20180081>) but is relying on an iridium catalyzed annulation reaction.

Another claim of the authors is, that they could overcome limitations of classical oxidative cyclodehydrogenation (typically using FeCl_3) by employing AlCl_3 in chlorobenzene at elevated temperatures, achieving full cyclization and planarization of both helical centers to yield the fully planarized structure DBTDI.

Additionally, the authors report high photoluminescence quantum yields (PLQY), with values up to 57% in solution and up to 32% in the solid state—which are claimed to be the highest reported for green light-emitting helical diamides.

2. Scientific Validity

The synthetic methodology is well-designed and clearly described. Structural assignments are convincingly supported by single-crystal X-ray diffraction. Photophysical properties are thoroughly characterized and discussed in the context of existing literature. The claims are well-supported by the data presented.

3. Significance and Novelty

The work is novel and represents an advance in the synthesis of helical and planar nanographene diimides. While the main building blocks and synthetic steps have been previously described, it has not been explored in this context. The angular fusion strategy and the use of AlCl_3 for full planarization are particularly noteworthy. The reported PLQY values are impressive and address a long-standing limitation in the field according to the authors.

4. Clarity and Presentation

The manuscript is generally well-written and clearly structured. However, the first figure in the introduction could be better aligned with the molecules discussed in the text to improve clarity. Additionally, the emphasis in the introduction appears to be more on planarization, whereas the true novelty lies in the generation of the helical structures. A slight rebalancing of this focus would enhance the narrative.

5. Specific Comments

Consider revising the first figure and its placement in the introduction to better reflect the molecular structures discussed.

A clearer emphasis on the novelty of the helical structure formation (rather than just the planarization) would strengthen the manuscript's impact.

Minor suggestions:

- 47: "Various synthetic strategies have been developed *to create h-NGs and planar structures*, such as photocyclizations, Scholl reactions, and cyclization's catalysed by transition metals."
I would advise to rewrite this sentence for clarity. I understand that there are synthetic strategies for the synthesis of polycyclic aromatic hydrocarbons in more general terms; or nano graphenes, independent if there are planar or helical. If the authors would like to refer to transformation of helical nanographenes into planar nanographenes by these synthetic strategies, it should be emphasized.
- 52: "In recent years, there have been several reports of incorporating six-membered dicarboximide *groups on periphery a helicene core, leveraging inspiration from the outstanding optoelectronic properties linked to rylene imides.*"
My suggestion is to revise this sentence to highlight the design approach using rylene imides as starting points and converting them into helical structures as it is outlined in the sentences afterwards.
- "52-71 and Figure 1"
Please number the structures in Figure 1 and refer to them in the text, for clarity and convenience of the reader.
- 106 "dry toluene, *uW*, 120 oC, 6h, 70%"
The abbreviation *uW* is not entirely clear to me from that context. It may refer to microwave?

6. Recommendation

Accept

The manuscript presents a novel and well-executed study that will be of interest to researchers in organic electronics, dye chemistry, and polycyclic aromatic hydrocarbons. The claims are well-supported, and the work meets the standards of Communications Chemistry.